# Predicting who responds to spinal manipulative therapy using a short-time frame methodology: Results from a 238-participant study

Maliheh Hadizadeh[1], Gregory Neil Kawchuk[1]*, Narasimha Prasad[2], Julie M. Fritz[3]

**1** Department of Physical Therapy, Faculty of Rehabilitation Medicine, University of Alberta, Edmonton, Alberta, Canada, **2** Department of Mathematical and Statistical Sciences, University of Alberta, Edmonton, Canada, **3** College of Health, University of Utah, Salt Lake City, Utah, United States of America

\* gkawchuk@ualberta.ca

## Abstract

### Background

Spinal manipulative therapy (SMT) is among the nonpharmacologic interventions that has been recommended in clinical guidelines for patients with low back pain, however, some patients appear to benefit substantially more from SMT than others. Several investigations have examined potential factors to modify patients' responses prior to SMT application. The objective of this study was to determine if the baseline prediction of SMT responders can be improved through the use of a restricted, non-pragmatic methodology, established variables of responder status, and newly developed physical measures observed to change with SMT.

### Materials and methods

We conducted a secondary analysis of a prior study that provided two applications of standardized SMT over a period of 1 week. After initial exploratory analysis, principal component analysis and optimal scaling analysis were used to reduce multicollinearity among predictors. A multiple logistic regression model was built using a forward Wald procedure to explore those baseline variables that could predict response status at 1-week reassessment.

### Results

Two hundred and thirty-eight participants completed the 1-week reassessment (age 40.0± 11.8 years; 59.7% female). Response to treatment was predicted by a model containing the following 8 variables: height, gender, neck or upper back pain, pain frequency in the past 6 months, the STarT Back Tool, patients' expectations about medication and strengthening exercises, and extension status. Our model had a sensitivity of 72.2% (95% CI, 58.1–83.1), specificity of 84.2% (95% CI, 78.0–89.0), a positive likelihood ratio of 4.6 (CI, 3.2–6.7), a negative likelihood ratio of 0.3 (CI, 0.2–0.5), and area under ROC curve, 0.79.

**Data Availability Statement:** All relevant data are within the paper and its Supporting information files.

**Funding:** This project was funded by the National Center for Complimentary and Integrative Health at the National Institutes of Health (1UH3AT009293–01). The funder had no role in study design, data collection and analysis, decision to publish, or preparation of the manuscript.

**Competing interests:** The authors have declared that no competing interests exist.

## Conclusion

It is possible to predict response to treatment before application of SMT in low back pain patients. Our model may benefit both patients and clinicians by reducing the time needed to re-evaluate an initial trial of care.

## Introduction

Spinal manipulative therapy (SMT) is among the nonpharmacologic interventions for low back pain (LBP) recommended as a second-line or adjunctive treatment option after exercise or cognitive behavioral therapy [1]. Spinal manipulative therapy is described as a high velocity, low amplitude force applied to the vertebral column most often by chiropractors [2]. Although recommended in clinical guidelines, some patients with LBP appear to benefit substantially more from SMT than others [3]. This observation has initiated several investigations that have examined potential factors to modify patients' responses prior to SMT application (Table 1).

Of these investigations, several have concluded that baseline characteristics can indeed be used to predict SMT response. A prospective study from the Nordic back pain subpopulation program examined 50 potential baseline factors in 875 LBP patients who received chiropractic care [24]. Their model correctly classified 99% of non-responders using 5 baseline variables: 1) sex, 2) social benefit, 3) severity of pain, 4) duration of continuous pain at first consultation, and 5) additional neck pain in the past year [24]. These results suggest that non-recovery from LBP in a chiropractic population is strongly related to demographic/self-report variables and weakly related to clinical variables; all five predictors were collected at the baseline without physical examination [24]. Interestingly though, the prediction rate for responders to chiropractic care was very low (6%). Further studies from this research group demonstrated similar results [12, 21]. Importantly, a subsequent validation study was performed by this group that constructed 5 predictive models on the basis of baseline information. None of the 5 models was sensitive (0–19%), whereas they were all reported highly specific (96–100%). Three factors were recognized as best at predicting non-responders by the fourth visit including no definite overall improvement by the second treatment session, the minimum total duration of LBP in the past year being 30 days, and presence of leg pain [18]. Similarly, a study using a pragmatic osteopathic approach that employed SMT found two statistically significant baseline variables including depression and pain intensity as predictors of back-related disability at 4 years [22]. Other studies from other groups have achieved similar results when consideration for symptom duration was given [14, 17].

Notably, a clinical prediction rule was developed to examine the characteristics of patient with LBP that may define a subgroup likely to benefit from SMT [23]. This work identified five predictive variables associated with 50% improvement in the Oswestry disability Index (ODI) within 1 week: duration of symptoms < 16 days, the fear avoidance beliefs questionnaire work subscale score < 19, at least one hip with > 35° of internal rotation range of motion, hypomobility in the lumbar spine, and no symptoms distal to the knee. According to this prospective, cohort study, patients were considered to be likely responders to manipulation when four or more of these variables were met. The probability of success with manipulation increased from 45% to 95%, when patients met this threshold. These predictive criteria was also investigated in a subsequent validation study [3]. The results showed LBP patients who received manipulation and met these criteria experienced greater decreases in pain and disability after 1, 4, and

**Table 1. The previous studies examined the predictive value of baseline variables for treatment outcome in patients with low back pain receiving SMT/chiropractic treatment.**

| Study/ Year of publication | Study population | Baseline sample size | Type of treatment | SMT technique | Number of SMT visits | Duration of SMT program | Response assessment time | Outcome variable/ Cut off value | Possibility of prediction | Study location |
|---|---|---|---|---|---|---|---|---|---|---|
| Eklund A et al. 2019 [4] | Patients with recurrent persistent LBP | 593 | Chiropractic treatment | Not reported | Not reported | Not reported | Fourth visit | Self-reported LBP status/ Definitely improved | Yes | Sweden |
| Eklund A et al. 2016 [5] | Patients with recurrent and persistent LBP | 666 | Chiropractic treatment | Not reported | Not reported | Not reported | Fourth visit | Self-reported LBP status/ Definitely improved | No | Sweden |
| Vavrek D et al. 2015 [6] | Patients with chronic LBP | 400 | SMT/ light massage + 5 min of hot pack treatment + 5 min of very low intensity pulsed ultrasound (0.5 watts/cm2) | Pragmatic | A dose of 0, 6, 12, or 18 SMT visits | 6-weeks | Shortly after completion of 6 weeks of care | $\geq 50\%$ improvement relative to the baseline pain intensity measured by the Modified Von Korff pain scale | No | U.S. |
| Field J et al. 2012 [7] | Patients with non-specific LBP | 404 | Not reported | Pragmatic | Not reported | Not reported | 14, 30 and 90 days following the initial consultation | PGIC and BQ/ Poor outcome was defined by a PGIC response of better or much better (score of < 6), a change in total BQ score of $\leq$46% and a change in pain ($\leq$ 2 points) and as derived from the pain sub-scale of the BQ | No | England |
| Peterson CK et al. 2012 [8] | Patients with acute and chronic LBP | 816 | Chiropractic treatment | Pragmatic | Pragmatic | Pragmatic | 1 week, 1 month, and 3 months after the start of treatment | The PGIC scale/ Patients responding better or much better (scores of 1 or 2) were categorized as "improved" and all other patients as "not improved." | Yes | Switzerland |
| Cecchi F et al. 2011 [9] | Patients with chronic LBP | 205 (SMT group: n = 69) | Booklet + advice to stay active + vertebral direct and indirect mobilization + SMT with associated soft tissue manipulation | Prescribed [10] | 4–6 SMT sessions (as needed) weekly sessions | 4–6 once-a-week sessions. 20 minutes each session (80–120 minutes of treatment altogether) | Discharge | LBP-related functional disability assessed by RMDQ (those who decreased their RM score <2.5 were considered non-responders) | No | Not reported |

(*Continued*)

**Table 1.** (Continued)

| Study/ Year of publication | Study population | Baseline sample size | Type of treatment | SMT technique | Number of SMT visits | Duration of SMT program | Response assessment time | Outcome variable/ Cut off value | Possibility of prediction | Study location |
|---|---|---|---|---|---|---|---|---|---|---|
| Field JR et al. 2010 [11] | New patients with LBP | 71 | Chiropractic treatment | Not reported | Not reported | Not reported | Second appointment, One month after the initial consultation | Scores > 5 on the PGIC were taken as improvement | Yes | Not reported |
| Leboeuf-yde C et al. 2009 [12] | Patients with LBP | 731 | Chiropractic treatment | Not reported | Not reported | Not reported | Fourth visit, 3 months | Self-reported LBP status/ Definitely better | No | Sweden |
| Malmqvist S et al. 2008 [13] | New patients with LBP | 984 | Chiropractic treatment | Not reported | Not reported | Not reported | Second and fourth visits | The outcome (global assessment of present status at the 4th visit) was defined as positive only for those patients who reported to be definitely better at the fourth visit (or at the last visit if treatment was ended before the fourth visit). | Yes | Finland |
| Langworthy JM et al. 2007 [14] | Patients with a new episode of non-specific LBP | 158 | Chiropractic treatment | Not reported | Not reported | Not reported | 6 weeks | Deyo's Core Set/ Not reported | Yes | UK |
| Underwood MR et al. 2007 [15] | Patients with LBP with a current episode duration of at least 4 weeks | 1116 | SMT SMT + exercise | Prescribed [16] | Eight sessions | 12 weeks | 3 months and 12 months following randomization | RMDQ score/ Not reported | No | UK |
| Newell D et al. 2007 [17] | Patients with LBP | 788 | Chiropractic treatment | Not reported | Not reported | Not reported | 4 and 12 weeks after the initial consultation | The BQ and PGIC scores/ Patients were categorised as 'better' if they chose the top two items of the scale | Yes | UK |
| Axén I et al. 2005 [18] | Patients with LBP | 1057 | Chiropractic treatment | Pragmatic | Pragmatic | Pragmatic | Fourth visit (or at the last visit if treatment was ended before the fourth visit) | Self-reported LBP status/ Definite improvement | Yes | Sweden |
| Axèn I et al. 2005 [19] | Patients with nonpersistent LBP | 674 | Chiropractic treatment | Pragmatic | Pragmatic | Pragmatic | Fourth visit | Self-reported LBP status/ Definitely improved | Yes | Sweden |

(*Continued*)

**Table 1.** (Continued)

| Study/ Year of publication | Study population | Baseline sample size | Type of treatment | SMT technique | Number of SMT visits | Duration of SMT program | Response assessment time | Outcome variable/ Cut off value | Possibility of prediction | Study location |
|---|---|---|---|---|---|---|---|---|---|---|
| Leboeuf-Yde C et al. 2005 [20] | Patients with LBP | 1054 | Chiropractic treatment | Pragmatic | Pragmatic | Pragmatic | Fourth visit | Self-reported LBP status/ Definitely improved | Yes | Sweden |
| Leboeuf-yde C et al. 2005 [21] | Patients with persistent LBP | 875 | Chiropractic treatment | Not reported | Not reported | Not reported | Fourth visit, 3 months and 12 months | Self-reported pain (a 0–10 box scale) and disability (the revised ODI)/ Improvement was defined as a reduction of 2 increments or more on the pain scale or as a 30% reduction in the pain score and a reduction of 20 points or more on the ODI or as a 30% reduction of the Oswestry score. | Not reported | Norway |
| Burton AK et al. 2004 [22] | Patients with LBP | 252 | Passive soft tissue stretching + passive articulation of the lumbar spine + SMT + positive encouragement + advice to stay active | Not reported | Mean = 6.6 sessions | Not reported | 4 years | RMDQ score/ A score of 0–2 on RMDQ was considered as recovered | Yes | England |
| Childs JD et al. 2004 [3] | Patients with LBP | 131 (SMT group: n = 70) | SMT+ exercise | Prescribed [23] | 2 sessions | 4 weeks | 1 week | ≥50% improvement in ODI | Yes | U.S. |
| Leboeuf-Yde C et al. 2004 [24] | Patients with persistent LBP | 875 | Chiropractic treatment | Pragmatic | Pragmatic | Pragmatic | Fourth visit, 3 and 12 months | Maximum pain score of 1/10 and a maximum ODI score of 15/100 | Yes | Norway |
| Axèn I et al. 2002 [25] | Patients with persistent LBP | 615 | Chiropractic treatment | Pragmatic | Pragmatic | Pragmatic | Fourth visit | Self-reported LBP status / Definitely improved | Yes | Sweden |
| Flynn T et al. 2002 [23] | Patients with LBP | 71 | SMT | Prescribed | 2 sessions | Treatment sessions were 2–4 days apart | Before the second and the third sessions | >50% improvement in ODI | Yes | U.S. |

(*Continued*)

**Table 1.** (Continued)

| Study/ Year of publication | Study population | Baseline sample size | Type of treatment | SMT technique | Number of SMT visits | Duration of SMT program | Response assessment time | Outcome variable/ Cut off value | Possibility of prediction | Study location |
|---|---|---|---|---|---|---|---|---|---|---|
| **Skargren EI et al.** 1998 [26] | Patients with low back or neck problems | 323 (chiropractic group: n = 179) | SMT, mobilization, traction, soft tissue treatment, instruction on individualized | Pragmatic | Mean sessions 4.9 (SD 2.0) | Mean 4.1 weeks (SD 3.3) | 12 months | Mean ODI score/ Not reported | Yes | Sweden |
| **Burton AK et al.** 1995 [27] | Patients with acute and subacute LBP | 252 | SMT+ Exercises + general advice | Not reported | Mean sessions 6.6 (SD 5.13) | Not reported | 12 months | RMDQ score/ Patients were considered recovered if they had a RMDQ score of 0–2 and not recovered if greater than 2. | Yes | England |

LBP: Low Back Pain, SMT: Spinal Manipulative Therapy, ODI: Oswestry Disability Index BQ: Bournemouth Questionnaire, PGIC: Patient Global Impression of Change, RMDQ: Roland Morris Disability Questionnaire

24 weeks compared to those who received manipulation but did not meet the criteria and those who met the criteria but did not receive manipulation.

On the contrary, a number of studies have had difficulty in identifying baseline characteristics of patients who respond to SMT. A secondary analysis of the large British randomized trial (UK BEAM) showed that patient baseline characteristics including age, work status, pain and disability, duration of episode, quality of life, and beliefs did not identify who was more likely to respond to manipulation or exercise with manipulation followed by exercise (combined treatment) [15]. Another retrospective analysis found that a lower baseline Roland Morris score predicted non-response to back school and individual physiotherapy but not to spinal manipulation which was provided over 4–6 weeks [9]. In another randomized controlled trial [6], researchers tried to build pre- and post- treatment models to predict responders to SMT and future pain intensity in 400 patients with chronic LBP. They reported the pre-treatment responder model in identifying SMT responders from their baseline characteristics didn't perform better than chance.

In addition, the predictive value of psychological factors in persons with LBP seeking help from chiropractors is uncertain. While an early study on the value of psychosocial variables with early identification of patients with poor prognosis showed initial psychosocial information in the form of the patient's cognitive coping strategies is highly predictive of the level of disability reported at 1 year [27], more recent studies have found little or no correlation with outcomes [5, 7, 11, 12, 14, 17].

Given the above, predicting SMT responder status at baseline may be confounded by several factors including the timeframe over which SMT applications are given, the use of additional interventions other than SMT, inclusion of treatment response variables and the choice of baseline characteristics. While many of these prior attempts at predicting SMT responder status are from pragmatic trials, application of SMT over longer time frames that reflect clinical practice may result in confounding with the natural history of the condition. Further, use of additional interventions found in clinical practice complicates interpretation and comparison between studies. Similarly, inclusion of treatment response variables voids the ability to make a baseline prediction. Finally, as our understanding of the predictive value of baseline

characteristics grows, choices of which characteristics are included or excluded in the final model can cause concern.

With these issues in mind, we conducted a secondary analysis of a prior study that provided two applications of standardized SMT over a period of 1 week. The design of this prior study provides a unique opportunity to mitigate many of the potential confounders described above. Specifically, the shortened time frame of this design increases the likelihood of observing responses arising solely from SMT while decreasing the possibility of including responses associated with longer term mechanisms (e.g. natural history, contextual effects) or additional intervention. We further benefit from this design as it employs a previously validated criteria to define SMT responders; improvement in self-reported ODI occurring over 2 treatment sessions [28]. Importantly, this criterion has been tied to improvements in physical measurements in responders including biomechanical, neurological and biological variables [29–31] that were also collected in this study and available for use in baseline predictions. The study design also includes other new variables that have not been used previously but are increasingly thought it influence outcome (e.g. lumbar spine stiffness measures [31–33], lumbar multifidus (LM) muscles contraction [30, 31, 34]).

Therefore, the objective of this study is to determine if the baseline prediction of SMT responders can be improved through the use of a restricted, non-pragmatic methodology, established variables of responder status, and newly developed physical measures observed to change with SMT.

## Materials and methods

### Primary protocol

In this current study, we performed a secondary analysis of data from a randomized controlled clinical trial. The original protocol for the primary study has been published previously [35]. In brief, the primary objective of the original study was to develop an optimized, multicomponent, SMT protocol using a phased, factorial design with three factors (additional SMT, multifidus muscle activation exercises, and spine mobilizing exercises). Sample size calculation was based on previous work in similar patient populations [31]. An initial sample of 280 participants was identified to provide at least 80% power to detect the minimum important differences for the patient-centered outcomes with a conservative 2-sided $\alpha = 0.025$ to account for co-primary outcomes. A more detailed explanation of sample size assumptions is provided in the protocol publication [35].

Participants for the original study were individuals between 18–60 years of age with a primary complaint of LBP with or without symptoms into one or both legs, and an Oswestry disability score of at least 20%. Potential participants were excluded if they were currently receiving mind-body or exercise treatment for LBP from a healthcare provider, had "red flags" for a serious spinal condition (e.g., spinal tumor, fracture, infectious disorder, osteoporosis, or other bone demineralizing condition, etc.), showed signs consistent with nerve root compression (diminished myotomal strength, muscle stretch reflexes or sensation, positive straight leg raise), were currently pregnant, or had prior surgery to the lumbosacral spine.

After initial screening, those who provided informed consent were enrolled in the study. Each participant completed forms related to personal demographics, clinical history, and patient-reported outcomes. One of the study clinicians then performed a baseline assessment to collect various physical measurements. All participants then received two separate sessions of SMT occurring one day to one week apart. Manipulations were provided by either licensed chiropractors or physical therapists associated with the study. Following SMT, a re-assessment

was conducted which collected the same baseline variables. Participants were categorized as SMT responders if their ODI score improved by 30% in 1-week reassessment.

The primary study received ethical approval from the University of Alberta (Pro00067152) and University of Utah (IRB_00092127) Institutional Review Boards. All the patients' data were fully anonymized. Permission to use anonymized data for the present study was obtained by the responsible authority, Julie M Fritz.

### Demographic and history measures

Basic demographic information including age, gender, race, ethnicity, weight, height, marital status, employment status, highest education level, and clinical history (e.g. duration of symptoms, comorbid health conditions, prior history of LBP) were collected.

### Patient reported outcome measures

Baseline assessment also included the ODI and Numeric Pain Rating Scale (NPRS) which were used as participant self-report measures of function and pain respectively [36, 37]. The Fear-Avoidance Beliefs Questionnaire (FABQ) was also collected to measure patient beliefs about how physical activity and work may affect their LBP and perceived risk for re-injury [38]. In addition, short forms from the University of Washington concerns about pain (UWCAP) and pain-related self-efficacy (UWPRSE) item banks were collected to measure the extent to which people catastrophize in response to pain and their degree of confidence in the ability to function with pain respectively. We also assessed the participant's risk of persistent disabling pain as low, medium, or high risk using the STarT Back Tool (SBT) [39]. Patients were asked about their expectations of LBP outcomes specifically related to medications, surgery, rest, X-ray, MRI, modalities, traction, manipulation, massage, strengthening, aerobic, and range of motion exercises.

### Physical examination measures

Physical examination measures included assessment of spinal (flexion, extension, left and right side-bending) [40] and hip range of motion (left and right internal rotation), lumbar segmental testing for mobility with manually applied posterior-anterior force [41], pain on palpation, straight leg raise (SLR) [42], Aberrant movements during lumbar range of motion [43], multifidus lift test at two levels (L4-L5 and L5-S1) and a prone instability test [42, 43].

### Instrumented measures

Both LM muscle activation and lumbar spine stiffness were evaluated at the baseline. Multifidus activation was measured with brightness-mode ultrasound images using a Sonosite Micro-Maxx (Sonosite Inc. Bothell, WA, USA) and a 60-mm, 2–5 MHz curvilinear array transducer based on a previously validated protocol [44]. Participants were positioned prone with their head neutral and a pillow under their abdomen to flatten the lordosis. Images were obtained at two vertebral levels (L4-L5 and L5-S1) in the parasagittal plane during rest (static) and submaximal contraction (dynamic) in response to the participant lifting a small weight with the contralateral hand. The weight was selected according to the participant's mass (<150 lb: 1.5 lb; 150-200 lb: 2 lb; and >200 lb: 3 lb). Three images were acquired in each state (relaxed and contracted) for each side and at two levels (L5/S1, L4/5), one side at a time. Images were stored and analyzed offline using ImageJ V1.38t software (National Institutes of Health, Bethesda, MD). Offline measures of LM thickness were obtained from determining the distance between the posterior-most aspect of the facet joint inferiorly and the plane between the multifidus and

thoracolumbar fascia superior for both the resting and contracted states. Multifidus muscle activation was calculated as: (Thickness $_{contracted}$−Thickness $_{relaxed}$) / Thickness $_{relaxed}$) [44]. The average of three measures was used for the analysis, for the total of 8 variables.

Lumbar spinal stiffness was assessed with the VerteTrack™ (VibeDx Corporation, Canada) which uses a rolling wheel system to apply vertical loads over the spine of a prone participant. The VerteTrack houses multiple sensors to provide continuous, real-time quantification of spinal deformation in response to a defined load. The resulting force displacement curves were used to calculate stiffness at each lumbar segment in N/mm. Terminal Stiffness was calculated as the ratio of the maximum applied force to the resultant displacement at each lumbar level [31]. Global stiffness was determined from the slope of force-displacement curve between 5 N and 60 N, representing the stiffness of underlying tissues throughout each trial [31]. One measure per lumbar segment corresponding to general stiffness, terminal stiffness, last load, and displacement were retained for analysis, for a total of 20 variables. The within- and between-session reliability and accuracy for spinal stiffness measures taken with this device has been evaluated previously [45, 46].

## Spinal manipulative therapy

All SMT sessions began with a brief assessment by the clinician to identify possible SMT contraindications. The preferred SMT technique has been described previously [3]. This procedure is performed with the participant supine. The clinician stands opposite the side to be manipulated and side-bended the participant. The side to be manipulated was the side identified as more painful on the basis of participant's report. If the participant couldn't identify a more painful side the clinician selected a side. The participant crossed their arms in front of the chest while the clinician rotated him/her and delivered a high-velocity, low-amplitude (HVLA) thrust to the anterior superior iliac spine in a posterior/inferior direction.

If this technique was not possible due to participant preference or comfort, a side-posture HVLA was performed. The participant laid on their uninvolved side with their superior leg bent to 90˚ and the clinician places their pisiform on to their posterior superior iliac spine and delivers a high velocity low amplitude (HLVA) thrust. Previous study found no difference in outcome between this SMT procedure and a side-posture HVLA technique [47] while both techniques have been found to be well-tolerated [47].

Spinal manipulative therapy was considered complete if a cavitation (i.e. a "pop") occurred following SMT application. If cavitation was not achieved, the participant was repositioned and SMT performed again. If no cavitation occurred on this second attempt, the clinician performed SMT on the opposite side. A maximum of 2 attempts per side was permitted. If no cavitation was noted after the fourth attempt, SMT was complete. The number of SMT attempts and the technique used were recorded by the clinician.

## Statistical analysis

All measures collected at baseline were used at the beginning of this analysis. Continuous data was summarized by means, medians and standard deviation. Categorical data was summarized by frequencies and percentages.

We have summarized the statistical methods used for data analysis in Fig 1. An initial exploratory analysis demonstrated that the collected variables at the baseline were associated with the relative changes in ODI. However, a high correlation was found between most of the ultrasound values, stiffness measures, and lumbar mobility testing results in bivariate correlation analysis (R $\geq$ ±0.7), therefore a principal component analysis using varimax rotation with Kaiser normalization was conducted to address this multicollinearity and reduce the number

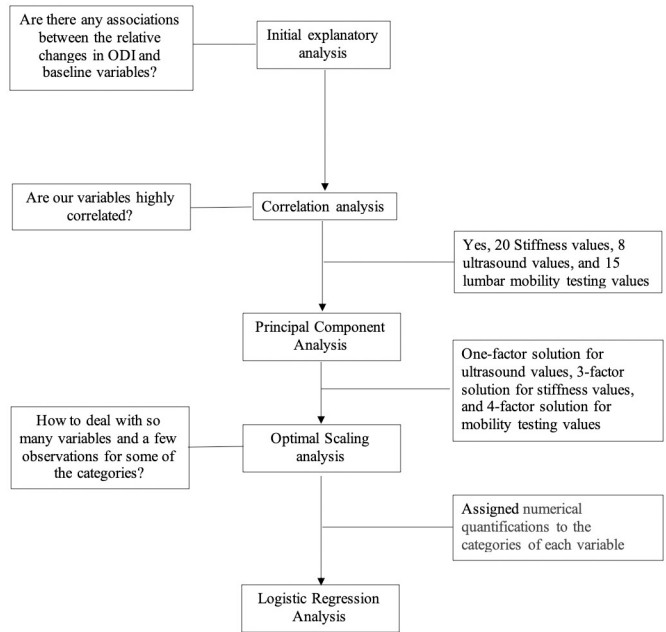

**Fig 1. Statistical analysis for prediction of response to spinal manipulative therapy.**

of variables input into the subsequent multiple regression model [15]. An optimal scaling analysis was also performed to address the problem of too few observations for some of the categorical variables. Optimal scaling is a general approach to treat multivariate data through the optimal transformation of qualitative scales to quantitative values. Using this approach, both nominal and ordinal variables can be optimally transformed into numerical values to reduce multicollinearity among predictors and maximize the homogeneity or internal consistency among variables. As a result nonlinear relationships between transformed variables can be modeled [48, 49]. Finally, a multiple logistic regression model was built using a forward Wald procedure to explore those baseline variables that could predict overall outcome (response status) at 1-week reassessment [6]. Analyses were conducted using IBM SPSS version 26.0 (Armonk, New York, USA). An alpha value of 0.05 was used for all analysis. In addition, sensitivity/specificity, positive/negative predictive values, positive/ negative likelihood ratios [50], and the area under the receiver operating characteristic (ROC) curve were estimated for the final model.

## Results

Two hundred and thirty-eight participants completed the 1-week reassessment (age 40.0± 11.8 years; 59.7% female). Tables 2–5 and 6 present the results of the history and demographic, patient-reported outcome measures, patients' expectations, physical examination and instrumented measures at the baseline, respectively.

Numeric pain rating scale reports the average of the worst, best, and current scores for pain over the last 24 hours using a self-reported 0–10 numerical pain rating scale ranging from '0' no pain, and '10' worst imaginable pain [37]. Function was evaluated using Oswestry Disability Index on a 0–100 scale, with lower numbers indicating better function [36]. Fear-avoidance beliefs about physical activity and work were assessed using the Fear Avoidance Beliefs Questionnaire (FABQ) [38]. The short form of the University of Washington concerns about pain (UWCAP) is a measure of pain catastrophizing including 8-items, with each item rated on a

**Table 2. History and demographic variables assessed at baseline.**

| Characteristics | All Participants (n = 238) | Responders (n = 68) | Non-responders (n = 170) |
|---|---|---|---|
| **Age (y)** | 40.0± 11.8 | 40.4± 10.8 | 39.8± 12.2 |
| **Sex (% female)** | 59.7 | 57.4 | 60.6 |
| **Race (%)** | | | |
| American Indian or Alaskan | 1.7 | 0.0 | 2.4 |
| Native | 10.5 | 11.8 | 10.0 |
| Asian | 2.9 | 4.4 | 2.4 |
| Black or African American | 73.1 | 64.7 | 76.5 |
| White or Caucasian | 6.3 | 13.2 | 3.5 |
| Other | 5.5 | 5.9 | 5.3 |
| > one race | | | |
| **Ethnicity (%)** | | | |
| Hispanic or Latino | 8.4 | 13.2 | 6.5 |
| Not Hispanic or Latino | 91.6 | 86.8 | 93.5 |
| **Marital status (%)** | | | |
| Single, widowed, or divorced | 36.6 | 30.9 | 38.8 |
| Married | 51.7 | 60.3 | 48.2 |
| Live with significant other | 11.8 | 8.8 | 12.9 |
| **Height (cm)** | 170.9± 10.4 | 168.9± 10.4 | 171.7± 10.4 |
| **Body mass index (kg/m2)** | 28.4± 7.0 | 27.5± 6.7 | 28.8± 7.1 |
| **Education level (%)** | | | |
| Did not complete high school | 2.1 | 1.5 | 2.4 |
| Completed high school | 34.9 | 23.5 | 39.4 |
| Completed college degree | 63.0 | 75.0 | 58.2 |
| **Current work status (%)** | | | |
| Not employed outside the home | 15.5 | 19.1 | 14.1 |
| Employed part-time | 17.2 | 16.2 | 17.6 |
| Employed full-time | 59.2 | 61.8 | 58.2 |
| Not employed for low back | 5.9 | 1.5 | 7.6 |
| condition | 2.1 | 1.5 | 2.4 |
| Retired | | | |
| **Workers' compensation (% yes)** | 3.4 | 4.4 | 2.9 |
| **Prior history of LBP (% yes)** | 61.8 | 60.3 | 62.4 |
| **Pain Duration** | 4000.0± 4149.0 | 3247.0± 3534.8 | 4301.1± 4343.7 |
| **Duration of current symptoms (d)** | 1116.5± 2312.4 | 1203.0± 2587.1 | 1082.0± 2200.3 |
| **LBP Frequency in the past 6 months (%)** | | | |
| Every day or nearly every day | 65.5 | 57.4 | 68.8 |
| At least half the days | 16.4 | 7.4 | 20.0 |
| Less than half the days | 18.1 | 35.3 | 11.2 |
| **Distal-most extent of symptoms (%)** | | | |
| Low back only | 41.2 | 38.2 | 42.4 |
| Buttock(s) | 37.4 | 48.5 | 32.9 |
| Thigh(s)—above the knee | 15.5 | 11.8 | 17.1 |
| Below the knee(s) | 5.9 | 1.5 | 7.6 |
| **Current medications regular usage for back pain (% yes)** | | | |
| Acetaminophen | 15.9 | 11.8 | 17.7 |
| Non-Steroidal Anti- | 26.5 | 19.2 | 20.4 |
| Inflammatories | 0.0 | 0.0 | 0.0 |

*(Continued)*

**Table 2.** (Continued)

| Characteristics | All Participants (n = 238) | Responders (n = 68) | Non-responders (n = 170) |
|---|---|---|---|
| Steroids | 6.7 | 4.4 | 7.7 |
| Opioid | 10.5 | 4.4 | 13.0 |
| Other | | | |
| **Comorbid health conditions (% yes)** | | | |
| Diabetes | 5.0 | 4.4 | 5.3 |
| High Blood Pressure | 8.0 | 7.4 | 8.2 |
| Cancer | 0.0 | 0.0 | 0.0 |
| Depression | 21.4 | 8.8 | 26.5 |
| Anxiety | 23.9 | 14.7 | 27.6 |
| Other mental health condition | 6.7 | 1.5 | 8.8 |
| Rheumatoid arthritis | 2.1 | 0.0 | 2.9 |
| Neck or upper back pain | 25.6 | 13.2 | 30.6 |
| Substance or alcohol abuse | 0.0 | 0.0 | 0.0 |
| **Cigarette Smoking history (%)** | | | |
| Non-smoker | 64.7 | 72.1 | 61.8 |
| Ex-smoker | 21.0 | 16.2 | 22.9 |
| Current smoker | 14.3 | 11.8 | 15.3 |
| **Previous tests (% yes)** | | | |
| X-rays | 57.1 | 45.6 | 61.8 |
| MRI | 27.3 | 22.1 | 29.4 |
| CT scan | 8.4 | 4.4 | 10.0 |
| Other imaging | 2.5 | 2.9 | 2.4 |
| None | 38.2 | 50.0 | 33.5 |
| **Treatment Used for LBP Episode (%yes)** | | | |
| Chiropractic | 46.6 | 35.3 | 51.2 |
| Physical Therapy | 40.8 | 30.9 | 44.7 |
| Steroid Injections | 13.4 | 8.8 | 15.3 |
| Corset/Brace | 8.4 | 10.3 | 7.6 |
| Opioid Medication | 19.7 | 14.7 | 21.8 |
| Massage Therapy | 37.8 | 26.5 | 42.4 |
| Cognitive Behavioral | 3.8 | 2.9 | 4.1 |
| Therapy/Counseling | 34 | 36.8 | 32.9 |
| Other | 20.6 | 29.4 | 17.1 |

NOTE. Values are mean ± SD unless otherwise indicated.

5-point scale: 1 (Never) to 5 (always). The higher the score, the more catastrophizing thoughts are present. The short form of the University of Washington pain-related self-efficacy (UWPRSE) was used to assess one's confidence in performing particular activities while in pain. It is a 9-item scale, with each item rated on a 5-point scale: 0 (Not at all) to 5 (very much). Higher scores represent higher confidence to function with pain. The short forms of the UWCAP and the UWPRSE items were scored by converting the total raw score into an item response theory-based T-score for with a mean of 50 and a standard deviation of 10. The mean score of 50 represents a mean of a large sample of people with chronic pain. The STarT Back Tool (SBT) is a 9-item questionnaire including physical and psychosocial statements that are used to categorize patients into low, medium, or high-risk groups for persistent LBP-related disability [39].

**Table 3. Patient-reported outcome measures at baseline.**

| Characteristics | All Participants (n = 238) | Responders (n = 68) | Non-responders (n = 170) |
|---|---|---|---|
| Numeric pain rating scale (0–10) | 4.6± 1.6 | 4.2± 1.7 | 4.8± 1.6 |
| Oswestry disability index (0–100) | 34.1± 11.8 | 34.0± 12.8 | 34.1± 11.4 |
| Psychosocial covariate measures | | | |
| Short form UWCAP | 49.2± 8.9 | 49.0± 8.1 | 50.5± 8.9 |
| Short form UWPRSE | 51.6± 8.2 | 53.3±7.5 | 50.9± 8.3 |
| FABQ score (0–96) | | | |
| Work subscale (0–42) | 15.6± 10.0 | 13.9± 9.1 | 16.3± 10.2 |
| Physical activity subscale (0–24) | 14.5± 4.9 | 14.0± 4.8 | 14.7± 4.9 |
| SBT total score | 4.3± 1.9 | 3.8± 1.8 | 4.6± 1.9 |
| SBT psychological distress score | 2.3± 1.4 | 2.03± 1.2 | 2.4± 1.4 |
| SBT categorization (%) | | | |
| Low risk | 33.2 | 44.1 | 28.8 |
| Medium risk | 46.2 | 45.6 | 46.5 |
| High risk | 20.6 | 10.3 | 24.7 |

NOTE. Values are mean ± SD unless otherwise indicated.

Principal component analysis identified a three-factor solution for the stiffness values, one-factor solution for ultrasound values, and four-factor solution for the mobility testing results. Together these factors explained 89.1%, 90.1%, and 78.3% of the variance in the stiffness, ultrasound, and lumbar mobility testing data respectively. Lumbar spine stiffness values, LM activation values, and mobility testing results were then converted into principal component scores to construct our model.

Logistic regression analysis resulted in a model with eight baseline variables (Table 7). The 8 variables in this model represent a number of different domains including participant demographics (height and gender), history (neck or upper back pain and pain frequency in the past 6 months), participant self-reported measures (SBT, patients' expectations about medication and strengthening exercises) and physical examination (extension status). Two variables were removed: One variable (depression) for not being statistically significant (P-value> 0.05) and another one (current pain duration) for having a regression coefficient of 0 and odds ratio (OR) equals to 1 showing there was no difference between responders and non-responders in the duration of their current pain.

As seen in Table 7, the effect of gender is significant but negative, indicating that females were 0.42 times less likely to respond to SMT than males. Higher expectations about strengthening (OR = 2.47) was associated with an increased likelihood of responding to SMT but higher expectation about medication (OR = 0.49) was associated with a reduction in the likelihood of responding to SMT. Participants with peripheralized pain during extension and those with more frequent pain in the past six month were 1.48 and 2.25 times more likely to be SMT responders, respectively. The ß coefficient for height, neck or upper back pain, and SBT score were also significant and negative indicating that increasing affluence is associated with decreased odds of responding to treatment.

Table 8 presents the degree to which predicted probabilities agree with actual outcomes in a classification table. The overall correct prediction, 81.5% shows an improvement over the chance level which is 50%. Our model had a sensitivity of 72.2% (95% CI, 58.1–83.1), specificity of 84.2% (95% CI, 78.0–89.0), a positive likelihood ratio of 4.6 (CI, 3.2–6.7), a negative likelihood ratio of 0.3 (CI, 0.2–0.5), and area under ROC curve, 0.79.

**Table 4. Patient expectations about different interventions at baseline.**

| Patients expectations (%) | All Participants (n = 238) | Responders (n = 68) | Non-responders (n = 170) |
|---|---|---|---|
| **Medications** | | | |
| **Completely disagree** | 10.9 | 13.2 | 10.0 |
| **Somewhat disagree** | 18.5 | 22.1 | 17.1 |
| **Neutral** | 24.4 | 33.8 | 20.6 |
| **Somewhat agree** | 42.0 | 26.5 | 48.2 |
| **Completely agree** | 4.2 | 4.4 | 4.1 |
| **Surgery** | | | |
| **Completely disagree** | 36.6 | 50.0 | 31.2 |
| **Somewhat disagree** | 18.9 | 14.7 | 20.6 |
| **Neutral** | 33.6 | 29.4 | 35.3 |
| **Somewhat agree** | 9.2 | 5.9 | 10.6 |
| **Completely agree** | 1.7 | 0.0 | 2.4 |
| **Rest** | | | |
| **Completely disagree** | 12.6 | 8.8 | 14.1 |
| **Somewhat disagree** | 11.8 | 10.3 | 12.4 |
| **Neutral** | 18.1 | 20.6 | 17.1 |
| **Somewhat agree** | 44.5 | 50.0 | 42.4 |
| **Completely agree** | 13.0 | 10.3 | 14.1 |
| **X-ray** | | | |
| **Completely disagree** | 16.4 | 16.2 | 16.5 |
| **Somewhat disagree** | 15.5 | 19.1 | 14.1 |
| **Neutral** | 40.3 | 35.3 | 42.4 |
| **Somewhat agree** | 19.3 | 20.6 | 18.8 |
| **Completely agree** | 8.4 | 8.8 | 8.2 |
| **MRI** | | | |
| **Completely disagree** | 11.8 | 13.2 | 11.2 |
| **Somewhat disagree** | 11.8 | 16.2 | 10.0 |
| **Neutral** | 37.8 | 33.8 | 39.4 |
| **Somewhat agree** | 29.0 | 27.9 | 29.4 |
| **Completely agree** | 9.7 | 8.8 | 10.0 |
| **Modalities** | | | |
| **Completely disagree** | 1.7 | 0.0 | 2.4 |
| **Somewhat disagree** | 2.9 | 2.9 | 2.9 |
| **Neutral** | 8.8 | 11.8 | 7.6 |
| **Somewhat agree** | 59.2 | 57.4 | 60.0 |
| **Completely agree** | 27.3 | 27.9 | 27.1 |
| **Traction** | | | |
| **Completely disagree** | 6.7 | 7.4 | 6.5 |
| **Somewhat disagree** | 3.8 | 4.4 | 3.5 |
| **Neutral** | 42.9 | 45.6 | 41.8 |
| **Somewhat agree** | 37.0 | 32.4 | 38.8 |
| **Completely agree** | 9.7 | 10.3 | 9.4 |
| **Manipulation** | | | |
| **Completely disagree** | 3.4 | 2.9 | 3.5 |
| **Somewhat disagree** | 4.2 | 5.9 | 3.5 |
| **Neutral** | 18.1 | 19.1 | 17.6 |
| **Somewhat agree** | 55.0 | 48.5 | 57.6 |

(*Continued*)

**Table 4.** (Continued)

| Patients expectations (%) | All Participants (n = 238) | Responders (n = 68) | Non-responders (n = 170) |
|---|---|---|---|
| Completely agree | 19.3 | 23.5 | 17.6 |
| **Massage** | | | |
| Completely disagree | 2.5 | 4.4 | 1.8 |
| Somewhat disagree | 3.8 | 0.0 | 5.3 |
| Neutral | 8.4 | 13.2 | 6.5 |
| Somewhat agree | 51.7 | 44.1 | 54.7 |
| Completely agree | 33.6 | 38.2 | 31.8 |
| **Strengthening exercises** | | | |
| Completely disagree | 0.8 | 0.0 | 1.2 |
| Somewhat disagree | 2.1 | 1.5 | 2.4 |
| Neutral | 6.3 | 4.4 | 7.1 |
| Somewhat agree | 39.1 | 30.9 | 42.4 |
| Completely agree | 51.7 | 63.2 | 47.1 |
| **Aerobic exercises** | | | |
| Completely disagree | 5.0 | 1.5 | 6.5 |
| Somewhat disagree | 10.9 | 11.8 | 10.6 |
| Neutral | 22.7 | 32.4 | 18.8 |
| Somewhat agree | 41.6 | 30.9 | 45.9 |
| Completely agree | 19.7 | 23.5 | 18.2 |
| **Range of motion exercises** | | | |
| Completely disagree | 0.8 | 1.5 | 0.6 |
| Somewhat disagree | 1.7 | 1.5 | 1.8 |
| Neutral | 7.6 | 5.9 | 8.2 |
| Somewhat agree | 42.0 | 38.2 | 43.5 |
| Completely agree | 47.9 | 52.9 | 45.9 |

## Discussion

Identification of SMT responders and non-responders prior to application of the SMT has received increasing attention in the conservative treatment of patients with LBP; however, the evidence for the effectiveness of this approach is mixed. To determine if the baseline prediction of SMT responders can be improved through the use of a restricted, non-pragmatic methodology, established definitions of responder status, and newly developed physical measures observed to change with SMT, we investigated the predictive values of 20 history and demographic variables, 6 patient-reported outcome measures, 22 physical measures, and 28 instrumented measures as unique domains and in combination. Our results suggest that it is possible to predict SMT response in a specific group of patients with 91.2% accuracy in non-responder and 57.4% in responder after only two applications of standardized SMT over a one-week period. To our knowledge, this is the first investigation to achieve prediction results of this magnitude for responder group although the model has yet to be validated.

Prior studies that have generated successful predictions of SMT response have tended to arise from pragmatic designs. In contrast, prior studies that have chosen to provide SMT alone or with minimal additional interventions have not achieved successful predictions. While it is possible that the prior success of pragmatic studies in this regard is because a pragmatic design more closely mimics clinical practice, our results do not support that idea. Specifically, our methodology applied fewer SMTs over a shorter time frame using a pre-defined technique for

**Table 5. Physical examination variables assessed at baseline.**

| Variable | All subjects (n = 238) | | | Responders (n = 68) | | | Non-responders (n = 170) | | |
|---|---|---|---|---|---|---|---|---|---|
| **Range of Motion** | | | | | | | | | |
| Right side-bending (°) | 25.4± 8.8 | | | 25.6± 8.7 | | | 25.3± 8.9 | | |
| Left side-bending (°) | 25.8± 8.9 | | | 26.8± 8.2 | | | 25.4± 9.1 | | |
| Total flexion (°) | 91.2± 24.3 | | | 95.2± 21.5 | | | 89.6± 25.3 | | |
| Total extension (°) | 24.5± 10.7 | | | 22.9± 10.5 | | | 25.1± 10.7 | | |
| Right hip internal rotation (°) | 31.0± 11.7 | | | 30.0± 11.9 | | | 31.4± 11.7 | | |
| Left hip internal rotation (°) | 31.0± 11.6 | | | 30.8± 11.8 | | | 31.1± 11.5 | | |
| **Right side-bending status (%)** | | | | | | | | | |
| Centralized | 16.8 | | | 10.3 | | | 19.4 | | |
| Status Quo | 76.5 | | | 86.8 | | | 72.4 | | |
| Peripheralized | 6.7 | | | 2.9 | | | 8.2 | | |
| **Left side-bending status (%)** | | | | | | | | | |
| Centralized | 12.6 | | | 5.9 | | | 15.3 | | |
| Status Quo | 80.7 | | | 89.7 | | | 77.1 | | |
| Peripheralized | 6.7 | | | 4.4 | | | 7.6 | | |
| **Total flexion status (%)** | | | | | | | | | |
| Centralized | 13.4 | | | 7.4 | | | 15.9 | | |
| Status Quo | 78.6 | | | 88.2 | | | 74.7 | | |
| Peripheralized | 8.0 | | | 4.4 | | | 9.4 | | |
| **Total extension status (%)** | | | | | | | | | |
| Centralized | 19.3 | | | 10.3 | | | 22.9 | | |
| Status Quo | 75.2 | | | 82.4 | | | 72.4 | | |
| Peripheralized | 5.5 | | | 7.4 | | | 4.7 | | |
| **Additional Tests** | | | | | | | | | |
| Right straight leg raise test (°) | 73.5± 14.5 | | | 72.9± 12.2 | | | 73.8± 15.4 | | |
| Left straight leg raise test (°) | 72.3± 16.1 | | | 73.3± 13.1 | | | 71.9± 17.1 | | |
| Aberrant movements during ROM (% Positive) | 37.4 | | | 45.6 | | | 34.1 | | |
| Multifidus lift test L4/L5 (% Abnormal) | 35.3 | | | 36.8 | | | 34.7 | | |
| Multifidus lift test L5/S1 (% Abnormal) | 39.5 | | | 45.6 | | | 37.1 | | |
| Prone instability test (% Positive) | 21.4 | | | 26.5 | | | 19.4 | | |
| **Manual Mobility Assessment (%)** | Hypomobile | Norm | Hypermobile | Hypomobile | Norm | Hypermobile | Hypomobile | Norm | Hypermobile |
| L1 mobility | 32.8 | 63.4 | 3.8 | 29.4 | 64.7 | 5.9 | 34.1 | 62.9 | 2.9 |
| L2 mobility | 34.0 | 61.8 | 4.2 | 32.4 | 61.8 | 5.9 | 34.7 | 61.8 | 3.5 |
| L3 mobility | 46.2 | 49.6 | 4.2 | 44.1 | 50.0 | 5.9 | 47.1 | 49.4 | 3.5 |
| L4 mobility | 58.8 | 36.1 | 5.0 | 60.3 | 36.8 | 2.9 | 58.2 | 35.9 | 5.9 |
| L5 mobility | 63.4 | 33.2 | 3.4 | 58.8 | 39.7 | 1.5 | 65.3 | 30.6 | 4.1 |
| **Pain on palpation (% yes)** | | | | | | | | | |
| L1 pain | 32.4 | | | 27.9 | | | 34.1 | | |
| L2 pain | 43.7 | | | 39.7 | | | 45.3 | | |
| L3 pain | 56.3 | | | 57.4 | | | 55.9 | | |
| L4 pain | 67.6 | | | 67.6 | | | 67.6 | | |
| L5 pain | 67.6 | | | 57.4 | | | 71.8 | | |

NOTE. Values are mean ± SD unless otherwise indicated.

**Table 6. Instrumented measures at baseline.**

| Characteristics | All Participants | | Responders | | Non-responders | |
|---|---|---|---|---|---|---|
| **Multifidus Activation** | | | | | | |
| Right L4_L5 | 3.8± 1.1 | | 3.8± 1.0 | | 3.8± 1.1 | |
| Left L4_L5 | 3.8± 1.1 | | 3.8± 1.1 | | 3.8± 1.1 | |
| Right L5_S1 | 3.6± 1.1 | | 3.6± 1.1 | | 3.6± 1.2 | |
| Left L5_S1 | 3.7± 1.2 | | 3.7± 1.2 | | 3.7± 1.1 | |
| **Multifidus Rest** | | | | | | |
| Right L4_L5 | 3.4± 1.1 | | 3.4± 1.1 | | 3.4± 1.1 | |
| Left L4_L5 | 3.5± 1.1 | | 3.5± 1.2 | | 3.5± 1.1 | |
| Right L5_S1 | 3.3± 1.2 | | 3.4± 1.2 | | 3.3± 1.2 | |
| Left L5_S1 | 3.4± 1.2 | | 3.5± 1.3 | | 3.4± 1.2 | |
| **Spinal Stiffness (N/mm)** | Global | Terminal | Global | Terminal | Global | Terminal |
| L1 | 4.5± 1.0 | 5.8± 1.1 | 4.6± 1.0 | 5.9± 1.1 | 4.5± 1.0 | 5.8± 1.2 |
| L2 | 4.4± 0.9 | 5.7± 1.1 | 4.5± 1.0 | 5.8± 1.1 | 4.4± 0.9 | 5.7± 1.1 |
| L3 | 4.4± 0.9 | 5.7± 1.1 | 4.6± 0.9 | 5.9± 1.1 | 4.4± 0.8 | 5.6± 1.0 |
| L4 | 4.5± 0.9 | 5.8± 1.2 | 4.7± 1.0 | 6.1± 1.2 | 4.5± 0.9 | 5.7± 1.1 |
| L5 | 4.7± 1.1 | 6.0± 1.3 | 4.9± 1.1 | 6.3± 1.4 | 4.6± 1.0 | 5.9± 1.3 |

NOTE. Values are mean ± SD.

**Table 7. Logistic regression analysis of 238 participants with low back pain for relative changes in Oswestry disability index following spinal manipulative therapy resulting in an 8-variable model.**

| Predictor | ß | Std. Error | Wald | P-Value | Odds ratio (e^ß) | 95% Confidence Interval | | Interpretation |
|---|---|---|---|---|---|---|---|---|
| | | | | | | Lower limit | Upper limit | |
| **Height** | -0.29 | 0.07 | 16.13 | 0.00 | 0.75 | 0.65 | 0.86 | Shorter, more improvement |
| **Gender** | -0.87 | 0.28 | 11.41 | 0.00 | 0.42 | 0.24 | 0.73 | Male, more improvement |
| **Current pain duration** | 0.00 | 0.00 | 6.35 | 0.01 | 1.00 | 1.00 | 1.00 | No changes |
| **Depression** | -0.39 | 0.22 | 3.32 | 0.07 | 0.68 | 0.44 | 1.03 | Not significant |
| **Neck or upper back pain** | -0.63 | 0.21 | 9.25 | 0.00 | 0.53 | 0.35 | 0.80 | No neck or upper back pain, more improvement |
| **Pain frequency in the past 6 months** | 0.81 | 0.18 | 20.23 | 0.00 | 2.25 | 1.58 | 3.20 | More pain frequency, more improvement |
| **Patient's expectation on medication** | -0.72 | 0.20 | 13.23 | 0.00 | 0.49 | 0.33 | 0.72 | Lower expectation, more improvement |
| **Patient's expectation on strengthening exercises** | 0.90 | 0.35 | 6.60 | 0.01 | 2.47 | 1.24 | 4.93 | Higher expectation, more improvement |
| **The STarT Back Tool** | -0.31 | 0.10 | 8.80 | 0.00 | 0.74 | 0.60 | 0.90 | Lower score, more improvement |
| **Extension status** | 0.39 | 0.18 | 4.77 | 0.03 | 1.48 | 1.04 | 2.11 | Peripheralized pain with extension, more improvement |

**Table 8. The observed and the predicted frequencies for responders and non-responders to spinal manipulative therapy by logistic regression for the final model with the cut off of 0.50.**

| | Predicted | | |
|---|---|---|---|
| **Observed** | **Non-responder** | **Responder** | **% Correct** |
| **Non-responder** | 155 | 15 | 91.2 |
| **Responder** | 29 | 39 | 57.4 |
| **Overall % correct** | | | 81.5 |

Note. Sensitivity = 39/ (39+ 15) % = 72.2%. Specificity = 155/ (155+29) % = 84.2%. Positive predictive value = 39/ (39+29) % = 57.4%. Negative predictive value = 155/ (155+15) % = 91.2%.

SMT application. Therefore, one explanation for our non-congruent results is that our hypothesis is tenable; that is, predicting SMT response is best assessed in a short-time frame and in isolation of other interventions.

In addition, the magnitude of our SMT responder prediction was substantial greater when compared to prior studies that have not exceeded 19% to date. In the clinical prediction rule developed by Flynn et al, SMT response was predicted with 100% in non-responders and 19% in responders. Although this previous model consisted of fewer variables (i.e. 5) that is presumably easier to manage, the prediction performance for responders was lower. While at first glance it may appear unwieldy to use an 8-variable model including a 9-item questionnaire in a future clinical situation, 7 of the 8 variables can be collected in advance of the examination. The remaining one variable can be collected by clinicians with relative ease and expediency (extension status). In addition, one fourth of the model presented in the study is about patients' expectations on treatment. Although previous studies showed illness beliefs and beliefs about rehabilitation make a significant contribution to the prediction of different rehabilitation outcome indicators, the reason for this association remains unexplained [51–56]. However, it would be worthwhile to address the power of treatment expectations in comparison to other psychosocial factors in this group of patients. Importantly, none of the clinical measures included in our final model involved newly described physical measures involving special equipment and training (ultrasonic evaluation of muscle contraction, evaluation of spinal stiffness evaluation with a mechanical device).

The strengths of our study include a multi-site design which would tend to mitigate the possibility of our results arising from a specific population. Although most previous studies used other measures as response criteria, we defined our response value as 30% improvement on the ODI which is an accepted threshold of change based on minimal clinically important difference scores for this questionnaire [57, 58]. Given this and considering the high sensitivity and specificity of our prediction results, we propose that a future validation study of this model is warranted. If found to be valid, these 8 variable models could provide clinicians with the opportunity to construct a more focused intervention plan after only 1 week of care. This would benefit both patients and clinicians by reducing more traditional re-evaluation periods of an initial trial of care that may extend into multiple weeks with many more treatment sessions.

As with all experiments, our study had limitations. First, our sample was heterogeneous in terms of pain duration. Although most participants in this study could be classified as having chronic LBP, our inclusion criteria were not limited to chronicity. Since the original primary study was designed to assess therapeutic effects in a wide range of participants, it did not restrict enrollment to a specific duration of low back pain. Therefore, the usability of the proposed model cannot be easily extrapolated to populations that may be highly homogeneous in pain duration. Second, we did not have a control group, thus these outcome data cannot be regarded as a clinical prediction rule, however, it can inform the professions of what might be important in patients' clinical assessment.

## Conclusion

The 8 variable model presented here was able to predict SMT response with a sensitivity of 72.2% a specificity of 84.2%, and an overall classification accuracy of 81.5%. Given these results, and that 7 model variables can be collected prior to clinician engagement, future validation of the model is warranted. Should the model be valid, it may benefit both patients and clinicians by reducing the time needed to re-evaluate an initial trial of care.

## Supporting information

**S1 File. Data necessary to replicate the analyses.**
(XLSX)

## Acknowledgments

The authors would like to thank Dr. Randy Vollrath, Dr. Shannon Wandler, Dr. Moe Gebara, Dr. Lacina Barsalou, Lynne Wong, and Erica Marr for their assistance during the data collection phase.

## Author Contributions

**Conceptualization:** Maliheh Hadizadeh, Gregory Neil Kawchuk, Julie M. Fritz.

**Formal analysis:** Maliheh Hadizadeh, Narasimha Prasad.

**Funding acquisition:** Gregory Neil Kawchuk, Julie M. Fritz.

**Methodology:** Maliheh Hadizadeh, Gregory Neil Kawchuk, Julie M. Fritz.

**Project administration:** Gregory Neil Kawchuk, Julie M. Fritz.

**Supervision:** Gregory Neil Kawchuk.

**Writing – original draft:** Maliheh Hadizadeh.

**Writing – review & editing:** Gregory Neil Kawchuk, Narasimha Prasad, Julie M. Fritz.

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
