## [Editor Report · Decision Letter 0]

17 Aug 2020

PONE-D-20-19523

Predicting who responds to spinal manipulative therapy using a short-time frame methodology: results from a 238-participant study.

PLOS ONE

Dear Dr. KAWCHUK,

Thank you for submitting your manuscript to PLOS ONE. After careful consideration, we feel that it has merit but does not fully meet PLOS ONE’s publication criteria as it currently stands. Therefore, we invite you to submit a revised version of the manuscript that addresses the points raised during the review process.

ACADEMIC EDITOR: 1. WHY THE RANGE OF AGE GROUP IS SO WIDE 2. WHICH LOW BACK PAIN SPECIFY ACUTE, SUB ACUTE, CHRONIC OR NON SPECIFIC 3. WHY RANGE OF MOTION OF SPINAL ROTATION IS NOT MEASURED 4. HOW SAMPLE SIZE IS CALCULATED 5. ONLY 1 TECHNIQUE OF SPINAL MANIPULATION IS USED EXPLAIN

We look forward to receiving your revised manuscript.

Kind regards,

Deepak Kumar

Academic Editor

PLOS ONE

Journal Requirements:

2. In the ethics statement in the manuscript and in the online submission form, please provide additional information about the patient records used in your retrospective study. Specifically, please ensure that you have discussed whether all data were fully anonymized before you accessed them and/or whether the IRB or ethics committee waived the requirement for informed consent. If patients provided informed written consent to have data from their medical records used in research, please include this information.

3.We note that you have indicated that data from this study are available upon request. PLOS only allows data to be available upon request if there are legal or ethical restrictions on sharing data publicly. For information on unacceptable data access restrictions, please see http://journals.plos.org/plosone/s/data-availability#loc-unacceptable-data-access-restrictions.
---

## [Author Response · Author response to Decision Letter 0]

24 Aug 2020

Many thanks for your valuable comments and suggestions. We have made changes to the manuscript based on these comments (please see below).

1. Why the range of age group is so wide? 

Thank you for this question. We will preface our answer by underlining that the submitted manuscript is a secondary analysis of data already collected from a primary RCT. The protocol for this study can be accessed here (Trials. 2018;19(1):306) and is referenced in our manuscript. We agree though that it is very valuable for the reader to be provided with more information regarding the decisions made in designing the primary study. With respect to age range, the intent of the study was to see if specific therapeutic combinations were more effective than others. In this respect, we know of no theoretical rationale for why some age groups would respond differently than others. Therefore the age range was as wide as allowable. As such, we have now included this explanation in our manuscript to provide a better explanation for the reader of the original study design.

2. Which low back pain specify acute, sub-acute, chronic or non-specific?

As above, the original primary study was designed to assess therapeutic effects in a wide range of participants. Therefore, the original primary study did not restrict enrollment to a specific duration of low back pain. The analysis of the primary data was designed to stratify results based on pain duration which was a consideration of the original design. As such, we have now included this explanation in our manuscript to better inform the reader.

3. Why range of motion of spinal rotation is not measured? 

The primary study was one that was the result of many prior studies that explored which objective measure were most responsive to the treatments under consideration. Therefore, the primary study was not designed to look for which measures might be responsive – those measures were already determined (muscle contraction viewed by ultrasound and spinal stiffness measured by indentation). Therefore, it was not appropriate to include non-responsive outcome measures, such as ROM, in the primary study. As there are many physical measures that were not included in the primary study for this reason (e.g. forward flexion), we have revised our manuscript to talk about which measures were chosen for the above reasons rather than provide a list of all the measures that were not included. 

4. How sample size is calculated?

The sample size calculation was described in our protocol paper (Trials. 2018;19(1):306) and we now included this in our revised manuscript for clarification. 

5. Only one technique of spinal manipulation is used explain.

As described in our above responses, the primary RCT was a culmination of many explorative studies. Not only did we identify which physical measures were responsive to the therapeutic interventions, we also showed that one specific technique was effective in providing changes to participant disability scores. As this approach has been repeated in several studies from our group (Flynn T 2002, Fritz JM 2011, Wong AY 2015, Childs JD 2004, Cleland JA 2009), it has become the standard intervention in our studies. We have now clarified this in our revised manuscript. 

Journal Requirements:

 The correction has been made.

2. In the ethics statement in the manuscript and in the online submission form, please provide additional information about the patient records used in your retrospective study. Specifically, please ensure that you have discussed whether all data were fully anonymized before you accessed them and/or whether the IRB or ethics committee waived the requirement for informed consent. If patients provided informed written consent to have data from their medical records used in research, please include this information.

The following statement was added to the manuscript: “All the patients’ data were fully anonymized. Permission to use anonymized data for the present study was obtained by the responsible authority, Julie M Fritz.”

3.We note that you have indicated that data from this study are available upon request. PLOS only allows data to be available upon request if there are legal or ethical restrictions on sharing data publicly. For information on unacceptable data access restrictions, please see http://journals.plos.org/plosone/s/data-availability#loc-unacceptable-data-access-restrictions.

We uploaded the anonymized data set as mentioned above.

---

## [Decision Letter · Decision Letter 1]

10 Nov 2020

Predicting who responds to spinal manipulative therapy using a short-time frame methodology: results from a 238-participant study.

PONE-D-20-19523R1

Dear Greg,

First, please allow me to apologise that I was unable to process this manuscript in a timely manner. We’re pleased to inform you that your manuscript has been judged scientifically suitable for publication and will be formally accepted for publication once it meets all outstanding technical requirements.

Kind regards,

Gustavo Machado, PhD

Academic Editor

PLOS ONE
---

## [Editor Report · Acceptance letter]

13 Nov 2020

PONE-D-20-19523R1 

Predicting who responds to spinal manipulative therapy using a short-time frame methodology: results from a 238-participant study. 

Dear Dr. Kawchuk:

I'm pleased to inform you that your manuscript has been deemed suitable for publication in PLOS ONE. Congratulations! Your manuscript is now with our production department. 

Kind regards, 

on behalf of

Dr. Gustavo de Carvalho Machado 

Academic Editor

PLOS ONE